# Distinguishing Nanoparticle Aggregation from Viscosity Changes in MPS/MSB Detection of Biomarkers

**DOI:** 10.3390/s22176690

**Published:** 2022-09-04

**Authors:** Dhrubo Jyoti, Scott W. Gordon-Wylie, Daniel B. Reeves, Keith D. Paulsen, John B. Weaver

**Affiliations:** 1Thayer School of Engineering, Dartmouth College, Hanover, NH 03755, USA; 2Fred Hutchinson Cancer Center, Seattle, WA 98109, USA; 3Department of Radiology, Dartmouth-Hitchcock Medical Center, Lebanon, NH 03756, USA

**Keywords:** molecular sensing, MPS, MSB, nanoparticle aggregation

## Abstract

Magnetic particle spectroscopy (MPS) in the Brownian relaxation regime, also termed magnetic spectroscopy of Brownian motion (MSB), can detect and quantitate very low, sub-nanomolar concentrations of molecular biomarkers. MPS/MSB uses the harmonics of the magnetization induced by a small, low-frequency oscillating magnetic field to provide quantitative information about the magnetic nanoparticles’ (mNPs’) microenvironment. A key application uses antibody-coated mNPs to produce biomarker-mediated aggregation that can be detected using MPS/MSB. However, relaxation changes can also be caused by viscosity changes. To address this challenge, we propose a metric that can distinguish between aggregation and viscosity. Viscosity changes scale the MPS/MSB harmonic ratios with a constant multiplier across all applied field frequencies. The change in viscosity is exactly equal to the multiplier with generality, avoiding the need to understand the signal explicitly. This simple scaling relationship is violated when particles aggregate. Instead, a separate multiplier must be used for each frequency. The standard deviation of the multipliers over frequency defines a metric isolating viscosity (zero standard deviation) from aggregation (non-zero standard deviation). It increases monotonically with biomarker concentration. We modeled aggregation and simulated the MPS/MSB signal changes resulting from aggregation and viscosity changes. MPS/MSB signal changes were also measured experimentally using 100 nm iron-oxide mNPs in solutions with different viscosities (modulated by glycerol concentration) and with different levels of aggregation (modulated by concanavalin A linker concentrations). Experimental and simulation results confirmed that viscosity changes produced small changes in the standard deviation and aggregation produced larger values of standard deviation. This work overcomes a key barrier to using MPS/MSB to detect biomarkers in vivo with variable tissue viscosity.

## 1. Introduction

The development of magnetic nanoparticle (mNP) methods for in vivo applications began with magnetic particle imaging (MPI) [1]. The critical feature of MPI is the very high sensitivity obtained by measuring the harmonics of the nanoparticle magnetization in an alternating magnetic field. That sensitivity on current imaging systems is approximately 1 nanogram [2] but has been estimated to be as low as 1 picogram for longer measurements [3]. MPI has produced remarkable results [2,4,5,6,7]; see [8] for a recent review. 

We adapted MPI detection methods with the goal of investigating the microenvironment in vivo. We used the magnetic particle spectroscopy (MPS) of larger mNPs, which relax via Brownian rotation: the physical rotation of the mNP in its microenvironment. We use the term ‘magnetic spectroscopy of nanoparticle Brownian motion’ (MSB) to emphasize the link to the microenvironment. MSB allowed us to explore many physical properties of the microenvironment: the number of mNPs present [9,10], temperature [11,12], viscosity [13,14], matrix rigidity [15,16], relaxation [17,18], and pH [19]. However, the application of MPS/MSB with the most potential may be the measurement of molecular concentrations [20,21]. MPS/MSB has the unique potential to measure concentrations with microscopic probes in vivo at depth over significant periods of time, up to days and weeks, until the probes are gradually resorbed.

Since there is great potential for measuring concentrations of biomarkers in vivo, there is a great deal of effort being devoted to developing corresponding methods. There are many ex vivo methods of measuring the number of specific molecules, ELISA [22] being the most important for many applications. In vivo optical methods [23] exist but are limited to a few millimeters depth, and MRI methods [24,25] have limited sensitivity. MSB could be used to measure cytokine and chemokine concentrations in cancer and molecular markers for muscle degradation in cardiac disease [26]. Monitoring cytokine and chemokine concentrations noninvasively also has important applications in the long list of diseases with an inflammatory component, e.g., Alzheimer’s disease [27], epilepsy [28], neurodegenerative diseases [29], obesity [30], stroke [31], mental illness [32], and traumatic brain injury [33]. In many cases, the ability to measure the concentration of a specific molecule at a specific anatomical location could enable significant improvements in the course of diagnosis and treatment.

The concentration of a targeted molecule is estimated by measuring the change in the relaxation of the mNPs present. The mNPs are coated with antibodies, aptamers, or other targeting molecules that bind different epitopes of the targeted molecule (Figure 1). The targeted molecule will then bind two or more mNPs together. When bound together, the mNPs are not able to rotate as freely, reducing the magnetization induced in the applied magnetic field. This can be thought of as increasing the frictional forces on the mNPs. The relaxation is reduced further as more mNPs are linked together, allowing the concentration to be estimated.

We required a model of nanoparticle aggregation in order to calculate the MSB signal. Models of mNP formation through coalescence exist, e.g., as a function of surface potential [34] but are generally numerical [35] and are related to formation rather than antigen–antibody linking [36]. Optical papers on molecular linking have been published [37,38], but they are experimental and lack theoretical support. These existing models of aggregation could also work, but for simplicity in this paper we built a simple model of mNP polymer formation and simulated the resulting MSB signal. Finally, we compared the simulation with in vitro data. Our main finding, that aggregation has an MSB signal distinct from that of viscosity changes, should hold for any model of aggregation where the mNP size distribution changes. Thus, the choice of an aggregation model is not critical.

Nanoparticle aggregation is an important active area of research in many areas of nanomedicine. Table 1 provides a small sampling of recent studies. In some contexts, such as the present work and MRI-guided drug delivery [39], aggregation is desirable. In other contexts, such as cancer hyperthermia treatment [40,41] and antimicrobial Ag nanoparticles [42], aggregation can be an impediment. Similar to the present work, antibody detection using the aggregation of functionalized Au nanoparticles has recently been demonstrated [43], but microscopy techniques were used, which are not ideal for in vivo applications like MPS/MSB is. Finally, functionalized mNPs in MPS-based bioassays have been developed for many years by several groups; for a recent review see [44]. These methods fundamentally use biomarker-mediated mNP aggregation for biomarker detection (Figure 1). While these methods have been established in vitro, an important challenge remaining for potential in vivo applications is to distinguish between biological viscosity changes and biomarker-mediated aggregation.

In prior MSB publications, in vitro binding experiments were performed while keeping all other confounding variables such as temperature and viscosity constant. There was no way to distinguish between aggregation and local viscosity changes; both increased the amplitude of the lower harmonic of the mNP signal relative to that of the higher harmonic. In this paper, we will introduce a new metric that remains at zero for viscosity changes and is non-zero and rapidly growing for aggregation. As far as we could find, no such technique currently exists in the literature. The new metric will be particularly useful in vivo to establish whether binding as opposed to local tissue viscosity change has occurred.

## 2. Methods

### 2.1. Modeling the Fraction of Bound mNPs after Biomarker-Induced Aggregation

We built a simple model of biomarker-induced aggregation from first principles and population balance [34,45] as follows. Consider the size distribution of functionalized nanoparticles ρ(V,N) as a function of hydrodynamic volume V and amount of biomarker N. Given an initial distribution $\rho_{0}\left(V\right)=\rho (V,0)$, we were interested in determining how the distribution changes as biomarker is added. We modeled each mNP as uniformly coated with many binding sites. Each binding site and the protein molecule it binds to is very small compared to the size of the mNP. The biomarker molecules mediate dimer and higher linear polymer formation, that is, mNP-B-mNP and so on, where B is a biomarker molecule. The probability that two mNPs bind should be proportional to the number of binding sites on each mNP. Therefore, that probability is proportional to the product of their surface areas. Assuming the mNP-linker solution reaches a steady-state, we proposed the model:(1)$$\begin{array}{c} \frac{d\rho (V,N)}{\mathrm{dN}}=-c_{1}\left(N\right)S\left(V\right)\rho \left(V,N\right)\int_{0}^{\infty }S\left(V^{\prime }\right)\;\rho \left(V^{\prime },N\right){\;\mathrm{dV}}^{\prime }\\ +c_{2}\left(N\right)\int_{0}^{\infty }\int_{0}^{\infty }S\left(V^{\prime }\right)\;\rho \left(V^{\prime },N\right)S\left(V^{\prime \prime }\right)\;\rho (V^{\prime \prime },N)\;\delta \left(V-V^{\prime }-V^{\prime \prime }\right){\mathrm{dV}}^{\prime \prime }{\mathrm{dV}}^{\prime } \end{array}$$.

This continuum limit model allows the easy calculation of predictions and should be accurate, since there are typically large numbers of mNPs and linkers. N corresponds to the number of biomarker molecules; we will precisely define N shortly. The first term corresponds to the number of mNPs lost to a given size V, and the second term corresponds to the number of mNPs gained to this size by binding two smaller mNPs. We included $c_{1}$ and $c_{2}$ as positive-definite functions of N independent of V necessary to balance the units. Proceeding with this general setup, we took the surface area as a function of volume to be
(2)$$S\left(V\right)=\;{\left(6V\sqrt{\pi }\right)}^{2/3}$$
where, for simplicity, we assumed the mNPs to be spherical and ignored geometry effects for polymers. The meaning of the quantity ρ that we solved for is as follows. In the limit of small dV, the number of mNPs with a hydrodynamic volume between V and V+dV is given by ρ(V,N) dV. For simplicity, we normalized the initial state $\int_{0}^{\infty }\rho \left(V,0\right)\;\mathrm{dV}=1$, so that ρ(V,N) dV is the number of mNPs as a fraction of the initial number of mNPs. We enforced the conservation of the total hydrodynamic volume,
(3)$$\int_{0}^{\infty }V^{\prime }\rho \left(V^{\prime },N\right){\;\mathrm{dV}}^{\prime }{=V}_{\mathrm{TOT}}\;$$
for all N. When we multiplied both sides of Equation (1) by V and then performed $\int_{0}^{\infty }\;\mathrm{dV}$, we found (since V and N are independent, the derivative can be brought outside the integral),
(4)$$c_{2\;}{=c}_{1}\;\frac{\int_{0}^{\infty }S\left(V^{\prime }\right)\;\rho \left(V^{\prime },N\right){\;\mathrm{dV}}^{\prime }\int_{0}^{\infty }V\;S\left(V\right)\;\rho \left(V,N\right)\;\mathrm{dV}}{\int_{0}^{\infty }V\;J\left(V,N\right)\;\mathrm{dV}}\;$$
where, for convenience, we defined
(5)$$J\left(V,N\right)=\int_{0}^{V}S\left(V^{\prime }\right)\;\rho \left(V^{\prime },N\right)\;S\left(V-V^{\prime }\right)\;\rho \left({V-V}^{\prime },N\right){\;\mathrm{dV}}^{\prime }.\;$$

Thus, the conservation equation provided us with a constraint on $c_{1}$ and $c_{2}$. We were therefore down to only one free parameter. We could state the differential equation as:(6)$$\frac{d\rho (V,N)}{\mathrm{dN}}=c(N)\left[-\;S\left(V\right)\;\rho \left(V,N\right)+\frac{\int_{0}^{\infty }V\;S\left(V\right)\;\rho \left(V,N\right)\mathrm{dV}}{\int_{0}^{\infty }V\;J\left(V,N\right)\;\mathrm{dV}}\;J\left(V,N\right)\right]\;$$
where we combined $\int_{0}^{\infty }S\left(V^{\prime }\right)\;\rho \left(V^{\prime },N\right){\;\mathrm{dV}}^{\prime }$ and $c_{1}$, naming their product simply as c. We assumed that every biomarker linker molecule added to the mNP solution creates an mNP-B-mNP bond, i.e., every linker is used and there are no free linkers in the solution. This assumption was justified because the chemical binding affinity tends to be extremely high for commonly used targeting molecules such as antibodies or aptamers. We defined N as the fraction of mNPs that are bound, which must be equal to the number of biomarker molecules. This provided us with the constraint equation:(7)$$N=1\;-\int_{2.5}^{7.5}\;\rho \left(V^{\prime },N\right){\;\mathrm{dV}}^{\prime }.\;$$.

The limits of integration are the range of the volumes of monomers (Figure 2a) appropriate for a typical narrow size distribution of unaggregated mNPs. We could generalize, but that is beyond the scope of this paper. We numerically solved for c(N) as follows. The strategy was to first re-write Equation (6) with a re-definition: $d\tilde{N\;}\equiv \;c(N)\;\mathrm{dN}$. Then, we solved for
$\rho (V,\tilde{N\;})$ and used the constraint Equation (7) as
$N=1-{ʃ}_{2.5}^{7.5}\rho \left(V^{\prime },\tilde{N\;}\right)$. Using
$c(N)=1/(\mathrm{dN}/d\tilde{N\;})$, we approximated $c\left(N\right)=0{.0533\;N}^{2}+0.0507\;N$, accurate from N=0 up to at least N=0.5. We thus arrived at a fully determined model with no free parameters, which was interesting. We finally solved Equation 6 for ρ(V,N) with this calculated c(N).

This is a first-order differential equation with some integral terms. It does not have analytical solutions in general. However, we could take advantage of the differential equation being first-order and integrate it numerically in a straightforward manner in small increments of dN. We computed the integrals inside the square brackets using a simple and intuitive Riemann sum method with sufficiently fine partitions.

### 2.2. Simulations of mNP Rotations Using the Effective Field Model

We calculated the experimentally measurable magnetic signal using the effective field model [46]:(8)$$\frac{\mathrm{dM}}{\mathrm{dt}}=-\frac{2M}{\tau_{B}}\;\left[1\;-\;\frac{\xi_{0}\sin\left(\omega t\right)}{\alpha_{e}\left(t\right)}\right].\;$$

This is a relatively simple but useful differential equation derived from the Fokker–Planck formalism, a general approach to finding the distribution of magnetizations at a certain temperature. It is correct to leading order assuming the applied field frequency ω is low enough. At each small timestep, we found the effective field $\alpha_{e}\left(t\right)$ by inverting the Langevin function, L(α)=coth(α)- 1/α=M(t). We obtained $\xi_{0}{=\mu H}_{0}{/k}_{B}T$, where μ is the mNP magnetic moment and $H_{0}$ is the applied field amplitude. The Brownian relaxation time,
(9)$$\tau_{B}=\frac{3\eta V}{k_{B}T}\;$$
depends on viscosity η and hydrodynamic volume V. We used a ratio of polynomials approximation [47] for the inverse Langevin function (0.1% maximum error). To find the total signal, we first calculated the signal $M_{i}\left(t\right)$ of each species i (monomer, dimer, trimer, etc.) with its own effective field equation. Then we summed them, weighted by the number $n_{i}$ of the species from the aggregation model,
(10)$$M_{\mathrm{TOT}}\left(t\right)=\overset{\infty }{\underset{i=1}{\sum }}{\;n}_{i}{\;M}_{i}\left(t\right).\;$$

The upper limit of the sum is formally infinity, but the higher polymers are eliminated by sedimentation. Sedimentation is complicated and depends on the viscosity of the fluid and the time allowed. Therefore, we truncated the sum at a reasonable index (the results below are for maximum i=10, i.e., up to decamer). Figure 3 shows the solutions for typical parameter values.

We were justified to calculate independent effective fields binned by mNP aggregate size and add the signals together because we had a dilute mNP solution. There was negligible interaction between the aggregate species. Typically, we had a 5 μg/mL or lower concentration solution, which according to the Micromod specifications, contained about ${3\;\times \;10}^{12}$ mNPs per ml. The average separation between aggregates would therefore be approximately d=1 µm or larger, so the average interaction energies were vanishingly small. Consider the dipole–dipole interaction and Zeeman energies,
(11)$$E_{\mathrm{dip}}=\frac{\mu_{0}m_{1}m_{2}}{{4\pi d}^{3}}\;;\;{\;E}_{\mathrm{Zee}\;}{=m}_{1}{\;H}_{0}.$$

We found that $E_{\mathrm{dip}}=2{.4\;\times \;10}^{-22}\;J$, whereas $E_{\mathrm{Zee}}=4{.9\;\times \;10}^{-19}\;J$ for the applied field $H_{0}=10\;\mathrm{mT}$. This shows that the $E_{\mathrm{dip}}$ is small compared to $E_{\mathrm{Zee}}$, and thus the coupling between mNP aggregate species could be ignored. We simulated the average net magnetizations for each aggregate species; the magnetization grew approximately as the square root of the number of cores. This set the μ for each species. Further discussion on the calculation of mNP magnetization is provided in Section 4.

### 2.3. Dimer-Only Model

In addition to the full aggregation model that we developed in this paper, we conducted comparisons with a dimer-only model. The latter represents a scenario wherein the mNPs are only allowed to form dimers; no trimers or higher polymers are formed. This model is trivial in the sense that it is characterized by a single number; the number of dimers is exactly equal to the number of biomarker molecules. In contrast, the full aggregation model is more complicated. It is characterized by the relative fractions of all polymers, which have to be solved by numerically solving our non-linear differential Equation (6). The dimer-only model serves as a helpful comparison.

### 2.4. In Vitro MPS/MSB Apparatus

The MSB signal is the magnetization produced by mNPs in an alternating magnetic field. The basic apparatus diagramed in Figure 4 comprises a simple, relatively inexpensive system that can be used for either in vitro or in vivo applications. A drive coil produces the applied field from the frequency supplied by the phase lock amplifier that is used to detect the signal. The pickup coil that detects the field from the magnetization and the MNP sample producing that magnetization are inside the drive coil.

The central problem is isolating the voltage produced directly in the pickup coil by the applied field, termed the “feed-through”, from the much smaller voltage produced by the magnetization. The first method, used in MPI to achieve very high sensitivity, is to measure the harmonics of the frequency of the applied field, which are at frequencies where there are no other signals. We also used two other methods: The first was placing a balancing coil in the applied field with no mNP sample in it to cancel the feed-through in the pickup coil. The second was to use a perpendicular orientation by rotating the pickup coil so that it was perpendicular to the applied field and implement a small static field to orient the magnetization toward the pickup coil during the applied field zero crossings.

The measured signal was the time-derivative of the changing magnetic field from the mNP magnetization. We used a ratio of harmonics because it canceled out the coil coupling factor and the number of mNPs, allowing us to focus on the physical properties of the microenvironment such as the temperature, viscosity, and aggregation. The methods presented are not limited to the use of the ratio of harmonics. These methods should function for any metric used.

The temperature of the sample was held constant at 31 °C by surrounding the sample with a water bath. Figure 4b,c shows the water being brought in and out of the sample chamber using plastic tubing.

### 2.5. Signal Scaling

The mNP dynamics are complex, leading to a complex signal that depends on the hydrodynamic size distribution as well as the size and magnetic properties of the magnetic cores. In general, closed-form formulas for signals are complicated if they even exist. However, there is a commonly used method in engineering, known as the Buckingham Pi theorem, which uses dimensional analysis to deduce power laws between all of the physical quantities that describe a system. In previous MSB publications, we used this method to deduce temperature and viscosity changes. The method can be summarized as follows. A function f exists such that the magnetization M of pure Brownian mNPs can be expressed as a function of two dimensionless variables,
(12)$$\frac{M(\omega ,H)}{M_{0}}=f\left({\omega \tau }_{B},\frac{M_{0}V_{c}H}{k_{B}T}\right)\;$$
where $M_{0}$, ω, $\tau_{B}$, $V_{c}$, H, $k_{B}$, and T are the saturation magnetization, driving field frequency, Brownian relaxation time, magnetic core volume, driving field amplitude, Boltzmann constant and temperature, respectively. A consequence of this is as follows. The ratio of harmonics of the mNP signal, which is just a function of the magnetization, can be “scaled” in frequency to deduce the change in relaxation times between a target and a control mNP sample, everything else being held constant. Similarly, we could scale in H to measure temperature changes.

We used signal scaling as a tool to measure many features of the microenvironment in which the mNPs resided. Scaling is able to represent a signal and all its complexity. It allows the signal to be represented accurately when a known set of state variables are invariant. If any of the state variables that are assumed to be invariant do change, scaling fails to accurately map one signal onto another. For example, consider two mNP samples, where the first is just a water–mNP solution and the other also contains a small amount of glycerol, which increases the viscosity. In this case, we could find a scaling multiplier that accurately maps the two signals onto each other. Formally, a multiplier ζ exists such that we could scale the target (tar) signal in frequency to overlap with the reference (ref) signal,
(13)$$M_{\mathrm{tar}}\left(\zeta \omega \right){=M}_{\mathrm{ref}}\left(\omega \right).\;$$

Since the Buckingham Pi theorem assures us that the signal M is a function of ${\omega \tau }_{B}$, scaling the signal in ω is equivalent to scaling it in $\tau_{B}$. Thus, we measured the ratio of the relaxation time of the target sample to that of the reference sample,
(14)$${\zeta =\tau }_{B}^{\mathrm{tar}}{/\tau }_{B}^{\mathrm{ref}},\;$$
which equals the ratio of the viscosity η of the target sample to that of the reference sample via Equation (9), assuming all other parameters, e.g., temperature T and field amplitude H, are identical between the samples.

On the other hand, biomarker-mediated mNP aggregation is more complicated because it changes the size distributions. The control and target signals cannot be accurately mapped onto each other using a frequency-scaling multiplier. Therefore, when scaling fails to map signals accurately onto each other, the size distribution rather than the viscosity has changed, indicating that antibody-mediated aggregation has occurred. We quantified this “scaling violation” in order to quantitate the aggregation.

### 2.6. In Vitro Reagents

Reagents were used as received. BNF (bionized nano ferrite) starch-coated mNPs were obtained from Micromod, Rostock, Germany. These mNPs were multicore particles consisting of aggregates of iron-oxide crystals embedded/coated in a matrix of hydroxyethyl starch. Glycerol, neat, was obtained from Fisher. All dilutions were made using PBS 1X (Corning). A stock glycerol solution (50% by weight in PBS 1X) was used to prepare subsequent glycerol dilutions by volume. The glycerol stock was used to increase the sample viscosity. The mNP linker molecule used was concanavalin A, isolated from jack bean and obtained from VWR. The simulation and experimental parameters are listed in Table 2.

### 2.7. Numerical Solution

The solutions below were calculated in Matlab. Care had to be taken so that dN and dV were sufficiently small to ensure accuracy. One way to check that a solution is accurate is to compare the solution with that of a finer dN and dV; if the solutions converge then they are fine enough. Another accuracy check is to confirm that the conservation equation above is satisfied for every N. Finally, it was important to make sure ρ reached zero at the edges of the parameter space, that is V → 0 and $V\;\to {\;V}_{\max}$, otherwise there would be truncation and inaccurate results.

## 3. Results

### 3.1. MNP Magnetic Signal

Figure 3 shows the calculated magnetization for parameters representing the mNPs that we generally use experimentally. The steady state is reached rapidly by the third period. There is a phase shift relative to the applied field (dotted line), and a square-like shape indicating the presence of higher harmonics. The 50% bound case has transitions that are less sharp than those of the unbound case. Aggregation increases the hydrodynamic volume, and hence the frictional (viscous) forces, smoothing the rotation of the mNPs.

### 3.2. Biomarker-Induced mNP Aggregation

Figure 2 shows the solution to Equation (6). We used a narrow initial distribution of unbound mNPs, a normal distribution with a mean V=5, and a standard deviation of 0.32. As the biomarker concentration was slowly increased, we observed the mNPs binding together and forming dimers and higher polymers.

Figure 5a,b shows the distribution of aggregates as a function of the number of bonds. We conducted a comparison with a dimer-only model and highlighted the differences. It was interesting that the number of dimers peaked at around 10% by N=0.4; the divergence from the dimer-only model was evident. The total number of biomarker molecules, obtained by summing over the polymers, had a roughly linear relationship with N; it was about 30% at N=0.5. Figure 5b shows the first two moments of the polymer length distribution. As expected, the moments increased faster for full aggregation compared to the dimer-only model.

### 3.3. Ratio of Harmonics and Scaling

We calculated the scaling multipliers that would map each point to the reference line (using spline interpolation). Figure 6 shows the data scaled by the mean scaling multiplier. The reference spectra were chosen to have a larger applied field frequency range because the target spectra needed to be scaled on top of it.

We were free to choose any frequencies we liked as long as there were no natural resonances in the in vitro apparatus, which occur occasionally. In such a case, we could easily identify and skip over that frequency regime. We visually estimated that the standard deviation of the dimer-only model was smaller than that of the full aggregation model. We also observed that the viscosity increase (square dashed line) had zero standard deviation. This showed that aggregation violated scaling whereas viscosity preserved scaling.

### 3.4. Aggregation Versus Viscosity Data

Finally, as shown in Figure 7, we compared our simulation results to the experimental data. We observed in panel (a) (gray diamond line) that the mean scaling multiplier increased linearly with the viscosity. At a 50% viscosity increase, the scaling multiplier was 1.5, in exact agreement with the theory. We also observed that the standard deviation of the scaling multiplier remained at zero for viscosity, again in exact agreement with the theory, but increased monotonically for the dimer-only and full aggregation models. Thus, the simulation results showed that scaling was conserved by viscosity and violated by aggregation. The increase in standard deviation accelerated for the aggregation model in contrast to the dimer-only model, where it plateaued. This could be a qualitative difference between dimer-only aggregation and full aggregation. At N=0.5, the dimer-only model yielded scaling multipliers that differed from the full aggregation model by about 29%. This showed that the two models produced distinct signals, and thus highlighted the importance of calculating the full aggregation model. Indeed, the experimental data resembled the full aggregation model more than it did the dimer-only model.

The simulation results in panel (a) were echoed in the experimental data presented in panel (b). We observed a monotonic increase for aggregation in both the mean scaling multiplier and the standard deviation. The growth in the standard deviation resembled its counterpart in the simulation. For viscosity, the growth in the mean scaling multiplier was approximately linear, though there was some fluctuation which could be attributed to experimental precision, as discussed later. Finally, and most importantly, the standard deviation of the scaling multiplier for viscosity remained close to zero. This was in sharp contrast with that for aggregation, which was large in comparison, at a mean scaling multiplier of around 1.3, and continued to grow rapidly. We did not expect the viscosity standard deviation to be mathematically zero as in the simulation since the experimental precision of the MSB apparatus is finite. In summary, the experimental data appeared to be in good agreement with the simulations.

## 4. Discussion

Estimating the magnetization of aggregates is complex. For mNPs with relatively large anisotropy energies, the alignment of the core magnetizations during binding is dominated by the size of the mNPs. For smaller mNPs, dipole–dipole forces favor the anti-alignment of the magnetizations, tending to align the easy axes. However, dipole–dipole forces are smaller than thermal forces for larger mNPs, producing a random alignment of the easy axes.

The process is made more complex by surface interactions when the mNPs are close. Surface friction, including Van der Waals forces, inhibits the rotation of the mNPs relative to each other. For smaller anisotropy energies, the easy axes orientation becomes less important. The net magnetization of the aggregate becomes closer to the sum of the magnetizations of the component mNPs. For the mNPs we used here, the anisotropy energies were high and the sizes were large. The anisotropy energies were high enough that there was minimal Neel relaxation [48]. We also assessed the Neel relaxation in these particles by immobilizing them in Merogel and found that the remnant magnetization was small, less than 5%, indicating that most if not all of the signal originated from Brownian rather than Neel relaxation.

The high anisotropy and mNP size make dipole–dipole forces comparable to thermal forces only when they are bound by linker molecules; in this regime, the surface friction inhibits rotation. Therefore, we used a random orientation of easy axes in our calculation. However, a wide Boltzmann distribution of easy axes angles for smaller mNPs would probably not produce significant change in the simulation results.

In our simulations, we could have reached a higher N, but for our purposes N=0.5 sufficed. Within the limits of a higher N, it is possible that other binding effects come into play, limiting the accuracy, e.g., the mNP aggregates dissociate back into monomers as the binding sites fill up with an excess of biomarker molecules. Another possibility is that there is too much aggregation. That is, large polymers form; so large (about a micron in diameter) that they drop out of solution. In this case, the magnetic signal reaches zero. In practice, however, we found that our model had good correspondence with the in vitro experimental data for a large parameter regime. For example, we observed the in vitro detection of concanavalin A for concentrations spanning more than an order of magnitude, from the 50 pM detection limit to about 2 nM. We encountered similar binding data and ranges for other molecules, such as mouse cytokine interleukin-6 [49].

We used standard hand pipetting to add a few microliters of glycerol stock to each 100 ul in vitro sample to increase the viscosity. This method has a significant error margin but sufficed for this paper. The method could be improved, e.g., by using more expensive pipettes or enlarging the sample size. We showed that the dipole–dipole interaction could be ignored for the dilute mNP concentrations that we used. The concentration would have to increase by about ten-fold for the interaction energy to become comparable to the Zeeman energy. In this highly concentration limit, our effective field model binned by aggregate size would no longer be accurate; we would have to perform a stochastic simulation with particle interactions, which would be complicated.

Our metric is sensitive to the size distribution of particles. Anytime there is a variation in size distribution between a control sample and a target sample, the scaling violation is evident in the signals. If the particle size distribution changes because of something other than biomarker-induced aggregation, it causes a “positive” result in our test. However, mNPs are generally stable in solution, though they do have a limited shelf life and slowly self-aggregate over months, even in cold storage and faster at room temperature. Nevertheless, this aggregation can be easily accounted for using a control sample in practical in vitro and in vivo experiments.

However, the other effect that might violate scaling is the absorption of mNPs by phagocytic immune cells. We expect that in this case, the absorbed mNPs will experience a different frictional force compared to the unabsorbed mNPs, effectively changing the distribution of particle sizes $n_{i}$ in Equation 10, just as aggregation would. Until this scenario is studied, the molecules we are able to target might be limited to those that increase in concentration with upregulated immune response, so that the two effects combine rather than cancel each other out. Fortunately, most important cytokines do increase with upregulated immune function. However, this issue should be studied in future research.

The application of these methods is not limited to the measurement of concentrations, as they can also address the many factors impacting aggregation. Charged mNPs and those with thin coatings such that magnetic dipole interactions are relevant can also be studied using the rates of aggregation.

## 5. Conclusions

MPS/MSB frequency scaling similar to the method demonstrated in this paper has already been used to detect several pain and inflammation biomarkers in vitro, such as the cytokine interleukin-6 and granzyme-B [49]. The model for biomarker-mediated mNP aggregation and the Langevin model of mNP dynamics can be used to calculate the signal changes resulting from mNP aggregation. Simulations and in vitro experimental data using Micromod 100 nm iron-oxide mNPs both showed that changes in viscosity have a simple scaling relationship that is constant with frequency. Further, aggregation introduces nonconstant scaling over frequency that can be used to indicate biomarker- -mediated aggregation. The nonconstant scaling represents a test for biomarker -mediated aggregation in variable-viscosity media. Without such a test, it is impossible to isolate aggregation from biological viscosity variations. The nonconstant scaling identified in this paper is the smoking gun for biomarker detection via targeted mNPs. In future work, we plan to address a separate issue: the simultaneous accurate measurements of local viscosity changes and mNP aggregation. The simultaneous measurement of two independent physical quantities is possible with MSB [11]. This is a harder challenge that is beyond the scope of the current work, though this study represents an important steppingstone in that direction.

## Figures and Tables

**Figure 1 sensors-22-06690-f001:**
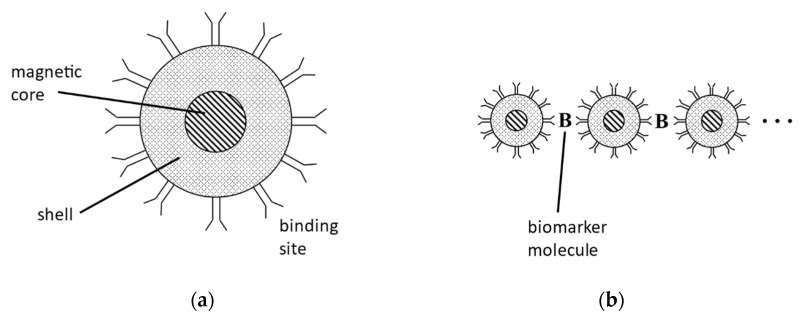
(**a**) Schematic diagram of an antibody-targeted magnetic nanoparticle (mNP). We typically use Micromod 100 nm hydrodynamic diameter nanoparticles with iron-oxide cores. A small applied oscillating magnetic field (~10 mT, ~1 kHz) produces a detectable response signal. Many binding sites are engineered to bind to a specific desired biomarker molecule. (**b**) Biomarker-mediated mNP aggregation. Formation of mNP-B-mNP-B-… polymers. The aggregated mNPs produce a magnetic signal that is distinguishable from that of the unaggregated state, allowing detection of the biomarker.

**Figure 2 sensors-22-06690-f002:**
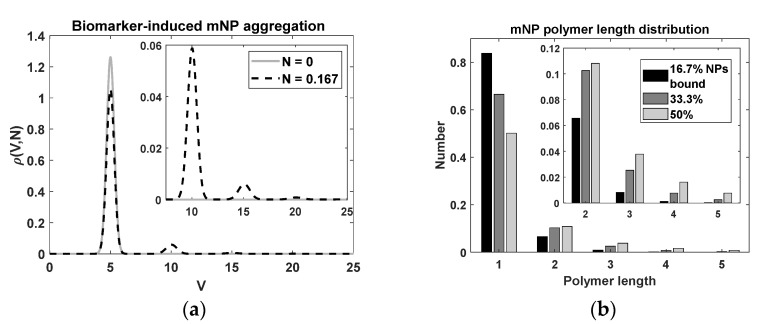
Model of biomarker-induced mNP aggregation. Insets show the same plots but zoomed in. (**a**) Initially (N=0), there were only unbound mNPs (solid gray line); by N=0.167 (dashed black line) a peak developed near V=10, indicating dimer formation. (**b**) The polymer length distributions for different N values up to N = 0.5. For example, if there were 100 mNPs, then at N=0.5 there were 50 unbound mNPs and about 11 dimers, 4 trimers, 2 tetramers, 1 pentamer, and so on.

**Figure 3 sensors-22-06690-f003:**
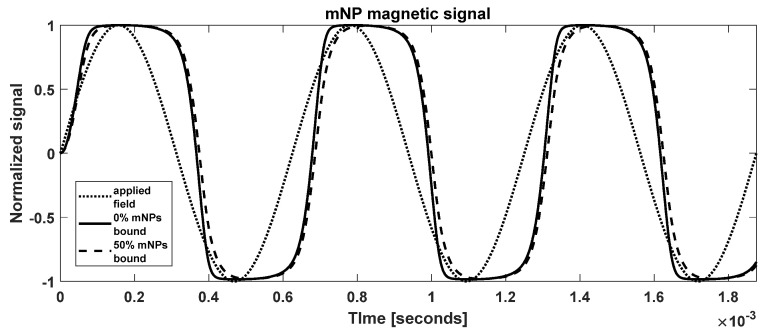
Simulated mNP magnetization from a sinusoidal applied magnetic field. Parameters are given in Table 2.

**Figure 4 sensors-22-06690-f004:**
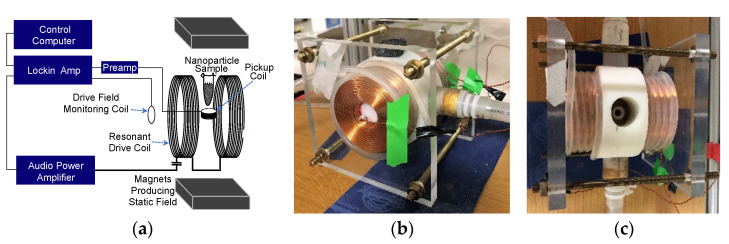
(**a**) Schematic of the MSB spectrometer used to obtain the mNP spectra. (**b**) The in vitro drive coil; outer diameter is about 3 inches. (**c**) Top view showing the chamber at the center of the coils where the nanoparticle solution 100 µL test tube sample is inserted for measurement.

**Figure 5 sensors-22-06690-f005:**
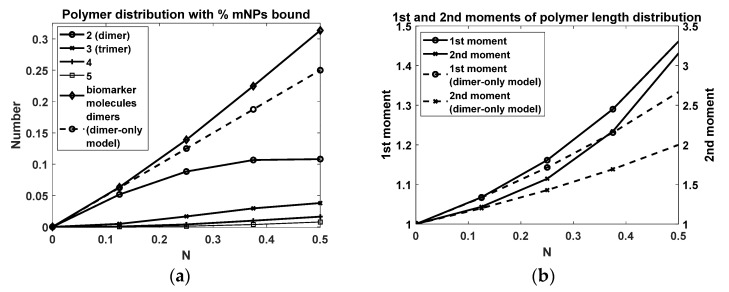
mNP polymer length distribution as a function of N (solid lines); comparison with dimer-only model (dashed lines). (**a**) Population of dimers through pentamers. (**b**) First moment (i.e., mean) and second moment.

**Figure 6 sensors-22-06690-f006:**
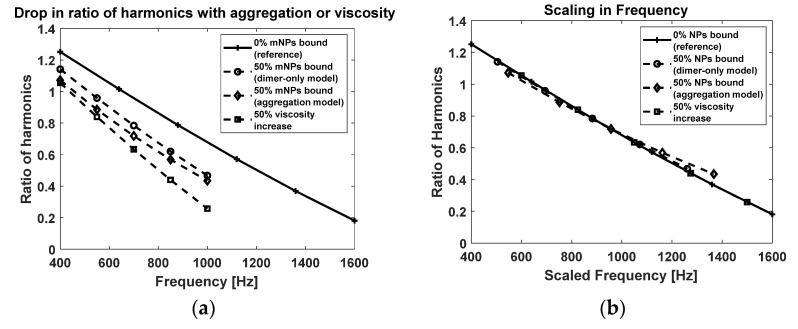
Magnetic spectroscopy of Brownian motion. Simulated harmonics of time-derivative of mNP magnetization. (**a**) Ratio of the fifth harmonic to the third harmonic decreased with increase in biomarker as well as viscosity. (**b**) Scaling in frequency to map the target signals onto the reference signal.

**Figure 7 sensors-22-06690-f007:**
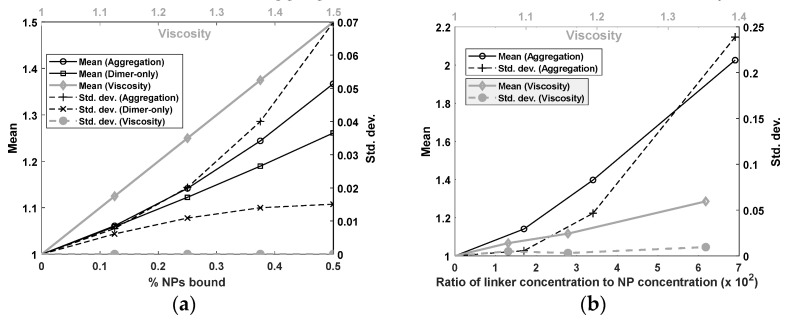
Comparison of dimer-only model, full aggregation model, and viscosity changes. (**a**) Simulation. (**b**) Experiment.

**Table 1 sensors-22-06690-t001:** Recent studies of nanoparticle (NP) aggregation in nanomedicine.

Study	Conclusion
MRI-guided NP drug delivery [39]	Fe_3_O_4_ NPs aggregate with stronger fields.
Cancer hyperthermia treatment [40,41]	Aggregation reduces heating.
Toxicity due to Ag NP aggregation [42]	Larger Ag NPs aggregate less.
Functionalized Au NPs [43]	Antibody detection with microscopy.
Bioassays with mag. spectroscopy [44]	In vitro detection of several biomarkers.

**Table 2 sensors-22-06690-t002:** Parameter values used for simulations and in vitro experiment.

Parameter	Value	Parameter	Value
mNP effective core diameter	15–25 nm	Temperature, T	31 °C
mNP hydrodynamic dia.	100 nm	Viscosity, η	0.78–1.17 mPa.s
Applied oscillating field, H	10 mT	Fe concentration	<5 µg/ml
Applied field frequencies, ω	0.4–2 kHz	mNP magnetization	49 Am^2^/kg Fe

## Data Availability

Not applicable.

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
