# Peer review of "Distinguishing Nanoparticle Aggregation from Viscosity Changes in MPS/MSB Detection of Biomarkers"

_sensors, 2022, doi:10.3390/s22176690_

Round 1

Reviewer 1 Report

The authors presented a numerical study of viscosity changes in MPS biomarker detection that enhance the aggregation of nanoparticles. The study is an interesting and well-written discussion. The results and the presented finding is interesting. However, the manuscript is not well prepared by the authors. The reviewer assumes this is the research article type of manuscript (not the review or communication, that all of them are indicated in the submission). 

Second, the affiliation seems incomplete without city, postcode, and country.

Please add the authors´ contributions after the conclusion.

What is the significance or the novelty of this model proposed by the authors to the field, compared to the existing model? a short review of the related published study and a summary in a comprehensive table will be a good additional discussion content for comparison and emphasize the new finding in this article.

More than 45% of the references in this submitted manuscript are self-citations, could the authors give any reason for this issue? 

The reviewer recommends the major revision and authors clarification for another round of review. 

Reviewer 2 Report

The authors have submitted the manuscript entitled “Distinguishing nanoparticle aggregation from viscosity changes in  MPS/MSB detection of biomarkers” has been submitted by the authors. Some issues to be addressed will improve the quality of the manuscript. Therefore, I recommend this work could be published after the minor revision.

1)    The background of this work is not clear. The authors should specify in a clearer way what novel and original this work propose to readers based on some new works. This research topic is widely studied in past and a lot of studies are performed. Author please added a comparative table for the reader’s clear understanding based on this study.

2)    The English composition requires many improvements. The authors should proofread the manuscript carefully to minimize grammatical errors.

Round 2

Reviewer 1 Report

 Accept in present form